# Grain Yields and Nitrogen Use Efficiencies in Different Types of Stay-Green Maize in Response to Nitrogen Fertilizer

**DOI:** 10.3390/plants9040474

**Published:** 2020-04-09

**Authors:** Wen Fu, Yang Wang, Youliang Ye, Shuai Zhen, Binghui Zhou, Yin Wang, Yujie Hu, Yanan Zhao, Yufang Huang

**Affiliations:** College of Resources and Environment, Henan Agricultural University, Zhengzhou 450002, China; fuwenchina@163.com (W.F.); zhen123nongda@gmail.com (S.Z.); zhoubinghui88888@gmail.com (B.Z.); wangyin981022@gmail.com (Y.W.); hu080369@gmail.com (Y.H.); zhaoyanan@henau.edu.cn (Y.Z.); huangyufang@henau.edu.cn (Y.H.)

**Keywords:** maize, senescence, nitrogen stress, stay-green, physiological characteristics

## Abstract

The stay-green leaf phenotype is typically associated with increased yields and improved stress resistance in maize breeding, due to higher nitrogen (N) nutrient levels that prolong greenness. The application of N fertilizer can regulate the N status of plants, and furthermore, impact the photosynthetic rates of leaves at the productive stage; however, N deficiencies and N excesses will reduce maize yields. Consequently, it is necessary to develop N fertilizer management strategies for different types of stay-green maize. For this study, the senescent cultivar Lianchuang 808 (LC808), moderate-stay-green cultivar Zhengdan 958 (ZD958), and over stay-green cultivar Denghai 685 (DH685) were selected as experimental models. Our results revealed that yields of ZD958 were slightly higher than DH685 and notably improved over than LC808. Compared with a non-stay-green cultivar LC808, ZD958 and DH685 still maintained higher chlorophyll contents and cell activities following the silking stage, while efficiently slowing the senescence rate. The supply of N fertilizer significantly prolonged leaf greenness and delayed senescence for ZD958 and DH685; however, the effect was not obvious for LC808. The stem remobilization efficiency of N was higher in the moderate-stay-green cultivar ZD958, in contrast to LC808, while the transfer of leaf N was lower than LC808, which guaranteed high leaf N levels, and that sufficient N was transferred to grains in ZD958. To obtain the highest yields, the optimal N fertilizer rates were 228.1 kg hm^−2^ for LC0808, 180 kg hm^−2^ for ZD958, and 203.8 kg hm^−2^ for DH685. In future, the selection of stay-green type crops might serve as an important agricultural strategy to reduce the quantity of N fertilizer and increase N efficiency.

## 1. Introduction

Maize (*Zea mays* L.) is one of the world’s most widely cultivated crops, providing food and animal feed as well as being a source of biofuel. [1,2]. Maize is predicted to become the first-ranked crop globally by 2020 [3]. The countries with the largest planted areas of maize are the USA, China and Brazil [4]. In the past century, maize grain yield has increased eight-fold with the majority of the yield being attributed to selection and hybrid breeding [5], which was due to increased maize greenness [6,7]. 

In 1987, Willman (1987) initially classified maize as non-stay-green and stay-green, contingent on the degree of the greenness of its leaves. The former category is also referred to as a premature senescence cultivar, which means that its leaves become less green 30 days after flowering, and then basically disappear at the grain maturity stage. The latter category refers to a sustained level of green (the overall proportion of a leaf that stays green) and no obvious loss of color at the maturity stage [8].

Early senescence translates to reduced green leaf area and photosynthesis, and significantly decreased grain yields [9]. While the chlorophyll and soluble protein content of stay-green maize leaves decrease slowly during senescence, the functionality of leaves is extended significantly [10,11] However, yields cannot be further increased, and are even reduced, when the leaf greenness score exceeds a certain threshold (over-stay-green), as much more N is retained in the leaves for photosynthesis, which does not transfer to grains, thus, reduce grain N concentration [12,13,14].

Further to N being the most essential nutrient for plant growth and development, it is also one of the nutritional elements with the greatest impact on grain yields [15,16]. Efficient N fertilizer management is vital to increase crop yields, improve soil fertility and minimize environmental risk [17]. As a critical component of chlorophyll, plant N levels are intimately related to leaf senescence and photosynthesis [18]. During the reproductive growth stage, N supplies prolong leaf greenness, while shortages of N induce early leaf senescence [19]. 

Leaf nitrogen (N) and photosynthesis are connected as most of the N in leaves is associated with photosynthetic machinery [20,21]. In contrast to a non-stay-green cultivar, the stay-green cultivar maintained more reduced nitrogen, chlorophyll content and higher nitrate reductase and carboxylase enzyme activities, which contributed to the accumulation of additional photosynthetic products during the grain-filling period [22]. In particular, the yield-increasing potential of the stay-green grain was more evident under the condition of N-deficiency stress [23]. 

To prolong leaf greenness following anthesis for the stay-green maize phenotype, more N was retained in the leaves, which influenced the accumulation of N in corn kernels. For senescent maize, the transfer of leaf N to reproductive organs is accompanied by rapid leaf senescence, which affects leaf CO_2_ fixation [14]. It follows that non-stay-green or stay-green cultivars have inherent advantages and disadvantages, where the application of N fertilizer might effectively regulate the N status of plants and further impact the process of leaf senescence. Therefore, we hypothesized that various stay-green types of maize should obtain optimized N fertilizer application rates to maintain leaf photosynthesis and N remobilization efficiencies. The objectives of this present study were to: (1) compare the senescence characteristics affected by different N application rates in various stay-green types of maize; and (2) determine the optimal N fertilizer rates for different stay-green types of maize.

## 2. Materials and Methods

### 2.1. Plant Material and Experimental Design

This study was conducted in Yuzhou City, Henan Province (34°27’ N, 113°34’ E), where the soil type of the test site was fluvo-aquic and clay. The pH of the cultivated soil layer was 7.44 with an organic matter content of 20.69 g·kg^−1^, which contained 0.96 g·kg^−1^ total nitrogen, 19.85 mg·kg^−1^ available phosphorus, 88.67 mg·kg^−1^ available potassium, and 58 kg hm^−2^ inorganic nitrogen (N_min_).

*Cultivar experiment*: In 2017, the present study was laid out in a randomized complete block design with three replications. A total of 20 maize cultivars (which are widely cultivated on the Huang-Huai-Hai plain) were used for the experiment, with the specific types of maize listed in Table 1. Seeds were mechanically sown on the 10th of June at a hill spacing of 0.60 m × 0.27 m, with 61,725 plants ha^−1^, with plot dimensions of 4 m × 10 m. Nitrogen (180 kg N hm^−2^) in the form of urea was applied in two splits with 50% at basal and 50% at the 10-leaf stage (45 d after sowing). Phosphorus [90 kg(P_2_O_5_) hm^−2^] in the form of calcium superphosphate, and potassium [90 kg(KCl) hm^−2^] were applied as a basal dose. The leaf SPAD value was measured at the silking and maturity stages, and the grain yield was weighed at harvest.

*N rate experiment*: In 2018 and 2019, the study samples were selected for cultivar experiments, including a senescent cultivar Lianchuang 808 (LC808), moderate-stay-green cultivar Zhengdan 958 (ZD958) and over-stay-green cultivar Denghai 685 (DH685), with the performance of the different maize cultivars in the field at maturity shown in Figure 1. Seeds were mechanically sown on June 08, at a hill spacing of 0.60 m × 0.27 m, with 61,725 plants hm^−2^, with the dimensions of each plot being 4 m × 10 m. The experimental design was a split plot with three replicates. three maize cultivars were applied to main plots, and five N treatment rates (0, 120, 180, 240 and 360 kg N hm^−2^) as the subplots. 

Urea served as the source of N, which was applied in two splits, with 50% at basal and 50% at the 10-leaf stage (45 d after sowing). Phosphorus [90 kg(P_2_O_5_) hm^−2^] in the form of calcium superphosphate, and potassium [90 kg(KCl) hm^−2^] were applied as a basal dose. The basal fertilizer was applied to the ground following manual broadcasting, whereas N topdressing was applied by means of side-dressing. Nicosulfuron and atrazine were applied at the three-leaf stage for weed control, whereas thiophanate-methyl and lambda-cyhalothrin were applied at the eight-leaf stage to prevent diseases and insect attack.

### 2.2. SPAD Values Measurements

The SPAD readings were obtained using a hand-held dual-wavelength meter (SPAD-502, chlorophyll meter, Minolta Camera Co., Ltd., Japan) from the mid-point of the ear leaf at the maize V6 stage (25 d after sowing), V12 stage (43 d after sowing), R1 stage (62 d after sowing), R3 stage (80 d after sowing) and R6 stage (107 d after sowing). The ear leaves of ten consecutive plants in one of the central rows were labeled with small plastic tags, and the SPAD values were measured in the morning (8:00–11:00 am). Finally, their average values are recorded as the values of the leaves.

### 2.3. Relative Leaf Conductivity

The ear leaves of 3 representative plants in each plot were selected as physiological test materials in the morning (10:00–12:00 am) at the silking stage and mature stage respectively, and the size and position of leaves in each ear position were consistent in the selection. To determine the relative leaf conductivity, 40 mL of distilled water was added to the leaves in a clean beaker, where after the conductivity R_0_ was measured using a conductivity meter (DDSJ 308, Shanghai). Five to eight well-grown leaves were removed from the top of the plant, which were then wiped with distilled water and dried, while avoiding the main veins. A hole punch Ø5 mm was used to obtain the mesophyll tissue (10 pieces each × 3). These samples were placed in a beaker, which was then sealed with plastic wrap and soaked for 5–6 h, after which the conductivity R_1_ was measured. Subsequently, the samples were placed in a water bath, boiled for 30 min, and then removed. After cooling to room temperature, the conductivity R_2_ was measured again. The relative conductivity R was calculated as: R = (R_1_ − R_0_)/(R_2_ − R_0_).

### 2.4. Chlorophyll Content

The ear leaves of 3 representative plants in each plot were selected as physiological test materials in the morning (10:00–12:00 am) at the silking stage and mature stage respectively, and the size and position of leaves in each ear position were consistent in the selection. The leaf disc samples were homogenized in 5 mL of an 80% acetone solution added to 0.01 g CaCO_3_, and then centrifuged at 2000× *g* for 10 min at 10 °C. The supernatant was collected, where the final volume of the extract was 25 mL using 80% acetone. The absorbance (A) of the extracts was determined at 470, 646.8 and 663.2 nm using a spectrophotometer, and an estimate of the chlorophyll content *a* [Chl *a* = 12.2.79 × A_646.8_], chlorophyll *b* [Chl *b* = 21.50 × A_646.8_−5.10 × A_663.2_], chlorophyll (*a* + *b*) [Chl (*a* + *b*) = 7.15 × A_663.2_ + 18.71 × A_646.8_] was obtained according to Lichthenthaler (1987) [24].

### 2.5. Plant Sampling and Determination of Total N Concentration

At maturity (107 d after sowing), five plants from each plot were sampled, and then dissected into leaves, stems, cobs and grains. These fresh materials were oven dried at 105 °C for 30 min and then at 75 °C, until a constant mass was achieved. The seeds harvested in each plot after sun-drying and threshing were weighed, The grain moisture of corn grains was adjusted to a 13%, which was recorded as the yield of the plot. and the samples in each plot were measured repeatedly for 3 times. The plant materials were ground to facilitate passage through a l-mm mesh screen, and then digested by H_2_SO_4_ and H_2_O_2_. The total N concentration of the digested samples was determined using an automated continuous flow analyzer (Seal, Norderstedt, Germany). 

### 2.6. Statistical Analyses

The formula for calculating the absorption and utilization efficiency parameters of the nitrogen fertilizer:TNA [kg hm^−2^] = plant N concentration [kg kg^−1^] × plant dry matter [kg hm^−2^];NRE [%] = (TNA of N applied − TNA of N omission) [kg hm^−2^]/N applied [kg hm^−2^] × 100 [%];N-PFP [kg kg^−1^] = Grain yield [kg hm^−2^]/N applied [kg hm^−2^];NHI [%] = grain N accumulation [kg hm^−2^]/TNA [kg hm^−2^] × 100 [%]
where TNA is the total N accumulation, NRE is the N recovery efficiency, N-PFP is the N partial factor productivity, NHI is the N harvest index.

A one-way analysis of variance (ANOVA) was applied to assess differences for each parameter using the Statistical Software Package for Social Science (SPSS, version 20.0). The mean values of the treatments were compared on the basis of the least significant difference test (LSD). The graphs were plotted using the Origin 9.0 software program. Linear regression was used to represent the relationships between yield and SPAD difference of the 20 maize cultivars. In the linear-plateau model, the point at which an increase in the independent variable no longer results in an increase in the dependent variable is termed the critical point or the critical rate. The inclined segment is described by the equation y = ax + b (if x critical point), and the horizontal segment is described by the equation y = c (if x > critical point), where a is the slope of the inclined segment and b and c are intercepts. SAS software (SAS 8.0, USA) was used to conduct the analyses and to obtain the relevant parameters.

## 3. Results

### 3.1. Relationship between △SPAD and Grain Yield

Fitting the difference between the SPAD of maize leaves at the silking and maturity stages (△SPAD) to the yield, revealed a relationship simulated by a quadratic functional relationship between the △SPAD and grain yield (y = 2.278x^2^ + 114.17x + 7809.4, R^2^ = 0.496; Figure 2). With higher △SPAD, the grain yield initially increased and then decreased. The yield attained its peak when △SPAD was 25, after which the yield began to decrease. This suggested that the proper stay-green increased the grain yields of maize; however, the over-stay-green reduced yields. 

### 3.2. SPAD Value 

With the growth of crops, the SPAD value of the different types of stay-green maize leaves was initially increased and then decreased, obtaining the maximum peak at the R1 stage (Figure 3). From the R1 to R6 stage (2018) the SPAD value of the leaves decreased significantly, with LC808 having the largest average decline (82.1%) followed by ZD958 (39.6%), where the lowest was DH685 (22.7%). Consistent with the SPAD reduction trend in 2018, the LC808, ZD958 and DH605 decreased by 84.8%, 43.5% and 22.8%, respectively, in 2019. Except for DH685 cultivar in 2018, there was no significant difference among treatments at jointing stage, but significant differences began to appear among treatments at the silking stage. With the increase of nitrogen application rate, SPAD increased, but when it exceeded 180 kg N hm^−2^, the increase of SPAD was not significant. At maturity, LC808 had no significant difference among treatments, and the treatment of 180–360 kg N hm^−2^ of other cultivars was significantly higher than that of no-N treatments.

In contrast to no-N treatments, the application of N fertilizer obviously increased the leaf SPAD value, where typically, DH685 exhibited the largest increase between the three cultivars. At the R6 stage, there was no significant difference in SPAD values between the different N levels of LC808, which were all <20. The ZD958 and DH685 were still significantly affected by N fertilizer; however, no obvious differences between the N_180_, N_240_, and N_360_ treatments were observed.

### 3.3. Chlorophyll Content

At the R1 stage, the application of N fertilizer significantly increased the total chlorophyll content in contrast to no-N treatments; however, no significant differences were observed between the different N addition treatments (except LC808 in 2018) (Figure 4). At the R6 stage, the total chlorophyll was increased with higher N application levels, and there were no significant differences observed at 180–360 kg N hm^−2^ loadings. Compared with the R1 stage, the chlorophyll content was significantly decreased at the R6 stage, where the reduction of Chl *a* content was the main driver behind the decline in total chlorophyll content. 

The differences in total chlorophyll content between the R1 and R6 stages were decreased with higher N applications. Moreover, the differences in chlorophyll content between the R1 and R6 stages were inconsistent for all cultivars. In contrast to the R1 stage, the mean total chlorophyll contents of LC808, ZD958, DH685 in 2018, were reduced by 9.1%, 25.5% and 14.6% respectively, and by 36.0%, 17.5% and 16.5%, respectively, in 2019.

### 3.4. Relative Electrical Conductivity (EC)

For the three maize cultivars, the relative EC of R1 stage was significantly less than that of the R6 stage, whereas the relative EC was reduced with the higher addition of N at the same growth stage (Figure 5). At the R1 stage, the average relative EC over all N treatments for the three cultivars was LC808 < ZD958 < DH685 (2018-19). However, the average relative EC value of the cultivars at the R6 stage showed the opposite trend in contrast to the R1 stage. At the R6 stage, the application of N fertilizer significantly reduced the relative EC values. The differences in relative EC between N_0_ and N_360_ treatments were 0.03 (LC808), 0.15 (ZD958) and 0.26 (DH685) in 2018, and 0.06 (LC808), 0.15 (ZD958) and 0.20 (DH685) in 2019. This indicated that the application of N fertilizer had the highest effects on the relative EC of DH685, with ZD958 second, and the poorest for LC808.

### 3.5. Grain Yield

The average maize yield in 2019 was 22.2% higher than that of 2018, which amounted to a significant difference in yields between these two years (Figure 6). The grain yield was increased with higher N rates, which attained a maximum when the quantity of N fertilizer was in the range of from 180–240 kg N hm^−2^. The N_360_ treatment yields were significantly decreased in contrast to N_240_ treatments in 2018 (except for cultivar LC808), whereas no significant decrease was observed in 2019. The yield of cultivar LC808 was less than that of ZD958 and DH685 for the two years. A regression equation revealed that the highest yields for LC808, ZD958 and DH685 were 7730.5, 8949.2 and 8775.6 kg hm^−2^, with corresponding N fertilizer rates at 228.1, 180.0 and 203.8 kg hm^−2^. Consequently, the moderate enhancement of the capacity of stay-green both increased grain yields and decreased the required N fertilizer dosage.

### 3.6. N Accumulation and N Use Efficiency

The accumulation of N in straw (stem and leaf) at the R1 stage was obviously higher than at the R6 stage, and stem N accumulation was significantly higher than the accumulation leaf N over the two years (Figure 7). The mean stem N transfer ratios across the different N treatments for the two years were 54.8% (LC808), 58.9% (ZD958) and 50.3%(DH685), while the mean leaf N transfer ratios were 63.5% (LC808), 47.8% (ZD958) and 41.0% (DH685). For the same cultivars, the stem N transfer ratio was decreased with higher N rates across all cultivars, and the leaf N transfer ratio was increased with higher N rates for LC808, while it was decreased with higher N rates for ZD958 and DH685.

The N recovery efficiency (NRE), N partial factor productivity (N-PFP) and N harvest index (NHI) were decreased under higher N application rates, and the effect of N on NHI was significantly lower than that of NRE and N-PFP (Table 2). These three related N use efficiency parameters were significantly different over the two years; the mean NRE and N-PFP in 2018 were less than that in 2019, whereas NHI showed the opposite trend. Furthermore, there were significant differences in N-PFP and NHI between cultivars, the N-PFP of the ZD958 and DH685 cultivars was higher than that of LC808, while the NHI of ZD958 was highest, followed by LC808, with DH685 being lowest.

## 4. Discussion

### 4.1. Stay-Green Improves Crop Yields 

Since the twentieth century, crop yields and stay-green scores have steadily increased, and have had a positive correlation [25]. By the 1970s, stay-green was the most important phenotype for breeding selection, particularly for maize [6]. Defects in chloroplast destruction or senescence-promoting mechanisms can cause leaves to retain their green colour during senescence, a phenomenon called “stay-green” [26]. The stay-green trait was verified to delay leaf senescence, but more importantly, it maintained leaf photosynthesis following the flowering period and increased grain yields [27]. Stay-green crops were classified into four types based on the differences of genes in leaf senescence and expression time [28]: type A, initiation of the entire senescence syndrome may be delayed; type B, the syndrome may begin on time but proceed at a decelerated rate; type C and type D were non-functional and had no value in production. Currently, the production of stay-green maize is primarily type B.

Regardless of type A or B, the delay in leaf senescence must be assured during the flowering to maturity period. Leaf senescence was primarily manifested as the leaf color changing from green to yellow until the whole leaf withered, where the internal mechanism was the reduction in chlorophyll content. As the SPAD value is a rapid and accurate method for representing chlorophyll [29,30], here, the classification of the stay-green scored in this study was based on the decrease in SPAD after flowering. Compared with the silking stage, the SPAD of LC808 at maturity was decreased by 82.1%, which was defined as a non-stay-green cultivar. The SPAD of ZD958 at maturity was decreased by 39.5% and defined as a stay-green cultivar, whereas the SPAD of LC808 at maturity was decreased by 22.7% and defined as an over-stay-green cultivar (Figure 3).

Plant cell membrane plays a role in regulating and controlling the exchange of substances inside and outside the cell, and its selective permeability is one of its most important functions. When the plant leaves are senescent or injured by stress, the cell membrane is damaged to varying degrees, the membrane permeability increases, the selective permeability is lost and the intracellular electrolyte exosmosis. The damage degree of membrane structure is related to the intensity of stress, the duration of stress, the resistance of crop varieties and other factors. Therefore, the determination of plasma membrane permeability can often be used as one of the physiological indexes of leaf activity [31,32,33,34]. In the present study, from the R1 to R6 stage, the mean relative EC increased by 170.0% in senescent LC808 maize, by 67.9% in the moderate-stay-green cultivar ZD958, and by 53.4% in the over-stay-green DH685 cultivar (Figure 5). This suggested that the degree of senescence for the stay-green cultivar was higher than that of the senescent type prior to the silking stage, whereas the rate of cell membrane damage of the senescent type was significantly higher than that of the stay-green type. 

The rapid rise of relative EC exacerbated the decomposition of Chl *a*; however, Chl *b* did not decrease much from silking to maturity (Figure 4). The concentration of chl *a* was higher than chl *b* at any point of time throughout the vase-life [35]. Compared with Chl *b*, Chl *a* also absorbed and transferred light energy, and a small portion of Chl *a* could convert light into electrical energy and produce chemical energy for dark reactions [36]. Research on rice also found that the Chl *a* degraded more quickly than did Chl *b* during aging [37]. However, whether of Chl *a* was significantly reduced required further study.

The retention of N in leaves maintained a long-term photosynthetic capacity [13]; however, it was observed that the NHI of ZD958 (moderate-stay-green cultivar) was higher than the other two cultivars (Table 1). This study found that a large quantity of N in the stem was transferred to the grains to compensate for N retention in the leaves of cultivar ZD958 (Figure 7). However, the NHI of DH685 (over-stay-green cultivar) was lowest between the three cultivars, where the insufficient efficiency of N transfer in leaves might have been the critical factor.

### 4.2. Effects of N Application on the Duration of Greenness for the Different Types of Stay-Green Maize 

The chlorophyll content of leaves can be significantly increased through the application of N, which facilitates the use of light energy by leaves, while improving the conversion of light energy to chemical energy [38,39]. Taylor et al. (2010) found that the sufficient application of N fertilizer can inhibit chlorophyll degradation, maintain cell viability, and prolong leaf greenness in plants. However, the effects of N fertilizer on delaying senescence and increasing the chlorophyll content of different stay-green types of maize were not clear [40]. 

At maturity, the leaves of the senescent cultivar LC808 at different N levels were underwent chlorosis. Leaf chlorosis of the moderate-stay-green cultivar ZD958 was observed only under insufficient N levels, the leaves maintained greenness under all treatments (Figure 1), and changes in the SPAD and chlorophyll content well reflected the changes in leaf color (Figure 3 and Figure 4). Additionally, the trend of relative EC was opposite to the chlorophyll content (Figure 5), which indicated that the N supply efficiently prolonged senescence for the stay-green and over-stay-green cultivars. 

It is worth noting that the addition of N fertilizer had a negligible impact on senescent cultivars, which limited their high-yield potential. In our study, the theoretical maximum yield of LC808 was 7730.5 kg hm^−2^, which was less than that of ZD958 (8949.2 kg hm^−2^), and DH685 (8775.6 kg hm^−2^), indicating it was difficult to a obtain high yield for the senescent cultivar (Figure 6).

### 4.3. Optimal N Rate for the Different Types of Stay-Green Maize

Insufficient or excessive N supplies can cause disorganized canopy structures that result in the degraded photosynthetic performance of crop populations, which can further reduce yields [41]. Our results demonstrated that leaf stay-green can improve maize yields; however, over-stay-green might reduce grain yields (Figure 2). As the plant N nutrient status was associated with the stay-green trait, dedicated N fertilizer management strategies should be specified for various types of stay-green maize. 

In the present study, the regression equation results revealed that the optimal N application rate (N_opt_) for LC808 = 228.1 kg hm^−2^, N_opt_ (ZD958) = 180.0 kg hm^−2^, and N_opt_ (DH685) = 203.8 kg hm^−2^ (Figure 6). Higher N absorption and N transfer efficiencies might be the main factors behind why stay-green cultivars do not require much fertilizer (Table 1). The highest attainable yield of over-stay-green cultivar was close to stay-green cultivar; however, the much higher levels of N retained in the straw could not transfer into grains, which resulted in higher N demands for the over-stay-green cultivar [12,42]. 

The extensive use of N fertilizers and the decline of N utilization efficiencies have caused widespread concern [43,44]. Big data has revealed that the maize N-PFP in China was only 25–37 kg kg^−1^; however, average global levels (44–72 kg kg^−1^) may be achieved though the improvement of N management strategies [45]. Cui et al. (2009) reported that the interaction between genotypes and nitrogen significantly influence grain yield and NUE in maize [46]. In the present study, the NRE and N-PFP was decreased with higher N application rates, and there were extremely significant differences between N levels or cultivars (Table 1). When the highest yields were achieved, the N-PFP of LC808 was ~37.5%, the N-PFP of ZD958 was ~53.2%, and the N-PFP of DH685 was ~49.4% (Figure 6; Table 2). China is the world’s largest producer and consumer of N fertilizers, which is projected to account for more than half of the global consumption by 2050 [47]. Thus, the selection of stay-green crops is an important strategy for China toward reducing its use of N fertilizers, while increasing their efficiencies.

Furthermore, in contrast to the senescent cultivar, stay-green maize contained more N in its straw [48]. Since most farmers in China return full straw to their fields, N fertilizer inputs should be appropriately reduced for the next crop season, in conjunction with the planting of stay-green (including over-stay-green) crops.

## 5. Conclusions

Moderate-stay-green ZD958 cultivar yields were slightly higher than the over-stay-green type DH685 cultivar, and greater than the senescent LC808 cultivar. The N fertilizer supply significantly prolonged leaf greenness and delayed senescence for ZD958 and DH685; however, this effect was not obvious for LC808. To obtain the highest yields, the optimal N fertilizer rate for LC0808 was 228.1 kg hm^−2^, 180.0 kg hm^−2^ for ZD958 and 203.8 kg hm^−2^ for DH685. The N remobilization efficiency from stems was higher in the moderate-stay-green cultivar ZD958 over LC808, while less leaf N was transferred than in LC808. The residual N in straw at harvest was much higher in ZD958 and DH685; therefore, if farmers return all straw to their fields, N fertilizer inputs should be adjusted (reduced) accordingly for the following growing season.

## Figures and Tables

**Figure 1 plants-09-00474-f001:**
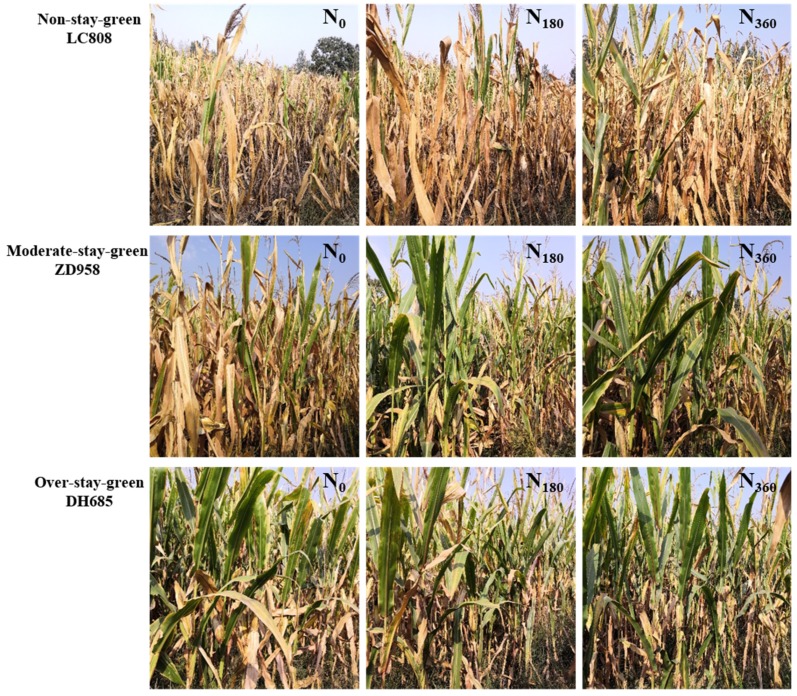
The performance of different stay-green types of maize under the different field nitrogen (N) levels at the maturation stage.

**Figure 2 plants-09-00474-f002:**
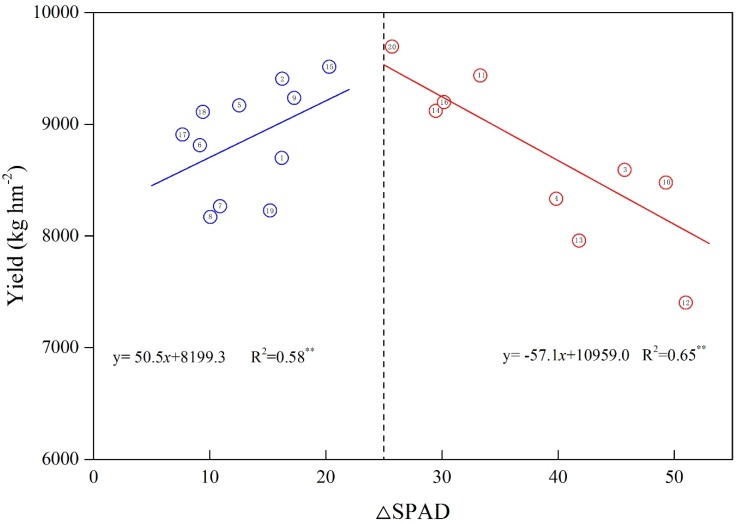
Relationships between the yield and SPAD difference of the 20 maize cultivars. The number in the circle represents the order of cultivar, and the name of cultivar refer to Table 1.

**Figure 3 plants-09-00474-f003:**
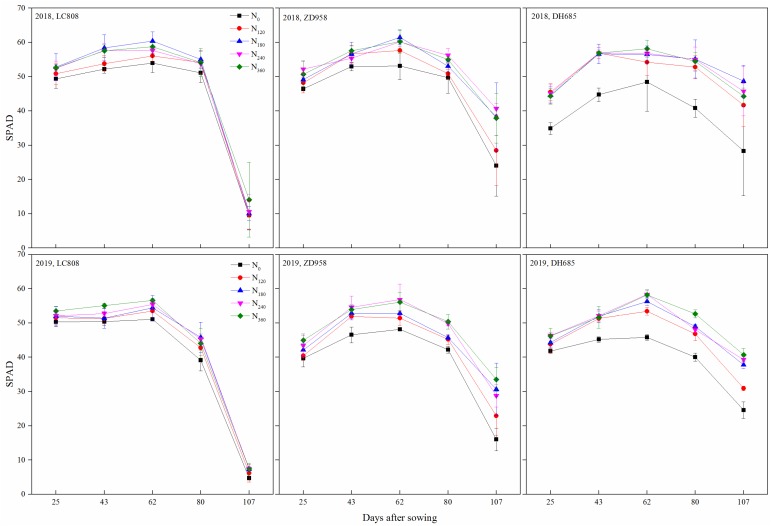
Dynamic of SPAD in different stay-green types of maize under the different N levels. Error bars indicate the SD.

**Figure 4 plants-09-00474-f004:**
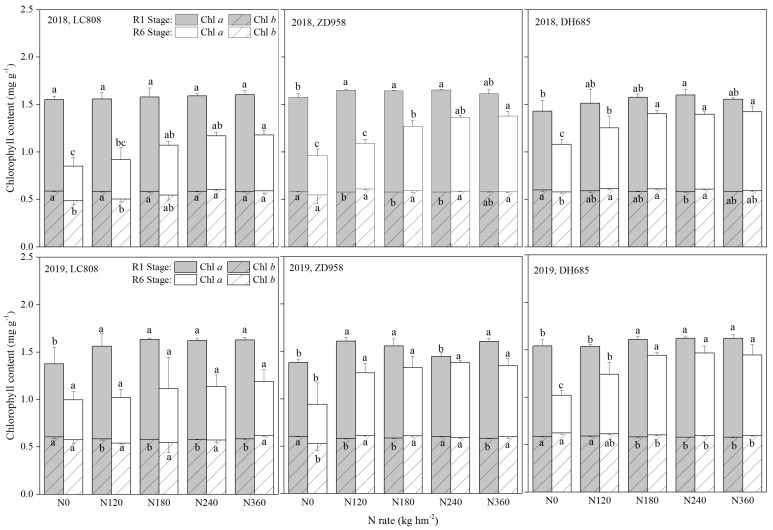
Chlorophyll content of different stay-green types of maize under the different N levels. Additional letters represent significant differences (*p* < 0.05) in Chl *a* or Chl *b* at the same growth stage between the different N rates. Error bars indicate the SD.

**Figure 5 plants-09-00474-f005:**
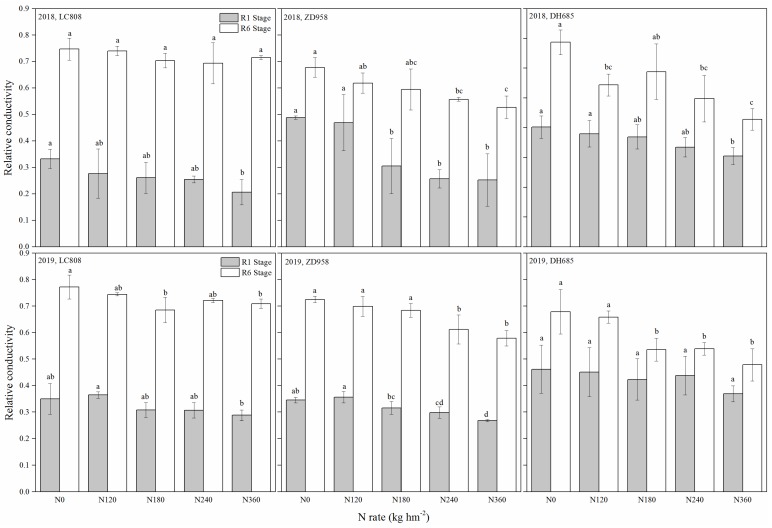
Relative conductivity of the different stay-green types of maize under different N levels. Additional letters represent significant differences (*p* < 0.05) at the same growth stage between the different N rates. Error bars indicate the SD.

**Figure 6 plants-09-00474-f006:**
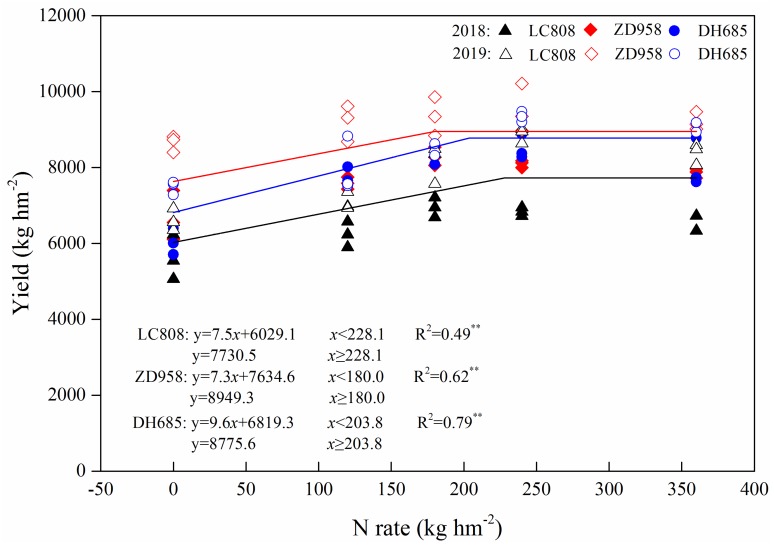
Regression equation between N fertilizer rates and yields for different types of stay-green maize.

**Figure 7 plants-09-00474-f007:**
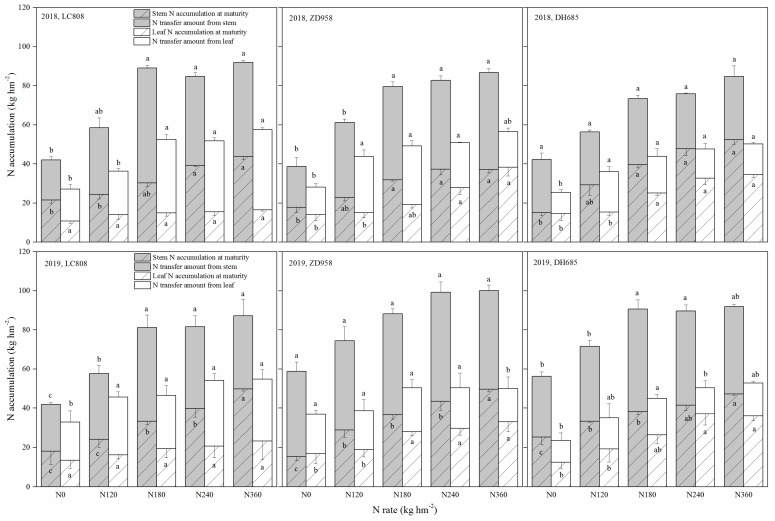
Effects of different N fertilizer application rates on N use efficiencies of different types of stay-green maize. Additional letters represent significant differences (*p* < 0.05) in N accumulation or N transfer amount for the same organ between the different N rates. Error bars indicate the SD.

**Table 1 plants-09-00474-t001:** The 20 selected maize cultivars.

No.	Cultivar	No.	Cultivar
1	Zhengdan958	11	Denghai939
2	Xianyu048	12	Fengdecunyu10
3	Lianchuang808	13	Lianchuang839
4	Cunyu10	14	Lianchuang825
5	Dika653	15	Denghai605
6	DikaJ1652	16	Denghai533
7	D4111	17	Denghai685
8	Dedan5	18	Xianyu1466
9	Denghai618	19	Xianyu1366
10	D4117	20	Xianyu335

**Table 2 plants-09-00474-t002:** Effects of different N fertilizer application rates on the N use efficiencies of different types of stay-green maize.

Cultivar	N Level	N Recovery Efficiency (%)	N PFP (kg kg^−1^)	N Harvest Index (%)
2018				
LC808	N_0_	-	-	65.2a
	N_120_	33.5a	57.0a	60.3a
	N_180_	34.0a	38.6b	65.7a
	N_240_	37.1a	26.0c	62.6a
	N_360_	22.3b	16.8d	62.1a
ZD958	N_0_	-	-	73.2a
	N_120_	38.6a	63.2a	76.4a
	N_180_	35.0a	44.8b	72.1a
	N_240_	34.7a	33.8c	70.0a
	N_360_	28.3a	21.5d	63.1b
DH685	N_0_	-	-	58.5b
	N_120_	47.9a	70.2a	66.6a
	N_180_	48.8a	51.4b	58.8b
	N_240_	47.3a	37.6c	58.2b
	N_360_	31.9b	23.6d	54.7b
2019				
LC808	N_0_	-	-	62.8a
	N_120_	48.6a	59.1a	63.3a
	N_180_	48.7a	45.2b	57.7b
	N_240_	41.4a	37.0c	58.8b
	N_360_	33.9b	24.1d	52.6b
ZD958	N_0_	-	-	64.2a
	N_120_	54.2a	76.7a	62.0ab
	N_180_	51.9a	53.8b	63.9a
	N_240_	46.1ab	39.6c	57.6b
	N_360_	33.6b	25.4d	60.3ab
DH685	N_0_	-	-	61.0a
	N_120_	55.5a	67.8a	60.8a
	N_180_	45.6ab	49.9b	54.8ab
	N_240_	38.0ab	38.9c	56.3ab
	N_360_	28.3b	26.2d	51.3b
*ANOVA*				
N rate (N)		14.01 **	546.17 **	0.47
Cultivar (C)		2.54	46.42 **	14.71 **
Year (Y)		14.48 **	43.24 **	18.25 **
N×C		0.34	2.99 *	2.58 *
N×Y		1.67	0.19	2.78*
C×Y		6.05 **	1.36	2.75
N×C×Y		0.49	1.49	1.17

Additional letters denote significant differences (*p* < 0.05) between the different N levels of the same maize cultivar in the same year. Values are mean ± SD (n = 3). * and ** indicate the variance with significance at the 0.05, and 0.01 level, respectively.

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
