# Peer review of "Grain Yields and Nitrogen Use Efficiencies in Different Types of Stay-Green Maize in Response to Nitrogen Fertilizer"

_plants, 2020, doi:10.3390/plants9040474_

Round 1

Reviewer 1 Report

The manuscript plants-753316 entitled “Grain yields and nitrogen use efficiencies in different types of stay-green maize in response to nitrogen fertilizer” is mature for publication. However, there are some minor points that in my opinion need improvement.

L.17, Compared with the non-stay-

L.26, “N fertilizer required”: not clear; “N fertilizer will be a prerequisite” ?

L.78, the proposed study: the present study ?

L.87, Table S1, No.15: 605 or 685 ?

L.98, maturation instead of mature

L.162: I would suggest you delete “that there was” and replace with “a relationship simulated by”

L.155: the approach does not sustain a two digit output, 25 would be enough.

Figure 2. I would suggest replacing dots with the numbers of the studied cultivars; this would be by far more useful. Moreover, I do not personally like such a trendline. I would divide the picture into two groups (1) below 25 and (2) above 25. I would draw a vertical line at 25 and then I would add linear treadlines, one ascending, one descending crossing each other somewhere near 25.

L.131: Plant sampling and determination of total N concentration

L.170, 172: DH605, please replace with the correct one.

L.191, L.264: What is the usefulness of the determination of the relative electrical conductivity. Please insert a paragraph, whereever you think is most suitable, to explain its utility, along with a reference.

L.273, “produce” instead of “synthesize”?

L.274, “the special element” ?, please delete.

L.295. Please eliminate the sentence “This was … senescence [32].”

L.297. “is” instead of “was”.

Author Response

Response to Reviewer 1 Comments

Reviewer 2 Report

It is my impression that the study presented is relevant and adds to the field.

The statistical methods reported seam to be adequate. However, in some passages, it is reported the results of some regression models that are not described in the material and methods (e.g., section 2.6), see for example the models used in section 3.1 (lines 150-155) or section 3.5 (lines 210-212). In section 3.2, please supply the significance of the comparisons used there.

The initial lines of section 2.6 (lines 138-143) should be edited. It is difficult to grasp what is reported there.

Author Response

Response to Reviewer 2 Comments

Reviewer 3 Report

Dear Authors,

The manuscript entitled " Grain Yields and Nitrogen Use Efficiencies in Different Types of Stay-Green Maize in Response to Nitrogen Fertilizer" focuses on comparing the senescence characteristics affected by different N application rates in various stay-green types of maize, and determining the optimal N fertilizer rates for different stay-green types of maize. The topic investigated and the approach proposed are innovative for the study area. The Title is clear and explicative. The Keywords proposed replicate words already used in the Title. I suggest checking them and use more specific words in order to better identify the topic of the study. The abstract is concise and well-written. The introduction informs the reader on the importance of the topic and key references are quoted. The aim of the study is clearly explained, and relevant scientific hypothesis added. In the materials and methods section, details related to experimental set-up, field measurement procedures and sampling times should be better described. Regarding data analyses, the statistical paragraph must be better detailed. I suggest to move the formulas reported in lines 139-144 in previous sections. The Results and Discussion sections are well organized, but findings must be better described and discussed also including key references on related studies carried out in similar environment and topic. The Tables and figures are informative, and they have a fully self-explanatory caption. In the conclusion section the Authors briefly summarized the main findings obtained, and future perspective presented.

Based on these arguments I recommend that this paper can be accepted for publication after minor revision.

Author Response

Response to Reviewer 3 Comments

Reviewer 4 Report

I have read the first paragraph of the introduction and concluded that further review of this paper is not warranted. There is no reason to believe that the scientific weaknesses seen in this paragraph do not permeate the rest of the paper. It will take a lot of time and effort for this voluntary reviewer to spell out all expected corrections. The authors should show from the very beginning that they master the art of scientific writing.

I recommend that the authors go through all references and make sure that they are all used and discussed in a proper manner.

Comments to the first paragraph of the introduction:

Lines 31–32: Reference [1] does not document the statement that “maize ... is he world’s most extensively produced crop”. The referenced paper deals with disease resistance in maize. The authors must refer to a publication providing data in support of the statement.

Lines 31–32: Reference [2] does not document the statement that “maize ... is he world’s most extensively produced crop”. The referenced paper deals with drought tolerance mechanisms in maize seedlings. The authors must refer to a publication providing data in support of the statement.

Lines 32–33: Reference [3] does not document the statement that “maize is predicted to become the world’s most important crop on a global scale by 2050”. The referenced paper deals with maize demand projections from 1995 to 2020. In addition, this is a secondary, popular publication – not a primary scientific publication.

Lines 34–35: Reference [4] does not support the statement that “The average production of maize in the United States increased from 3.6 Mg hm-2 in 1931 to 10 Mg hm-2 in 2010”. The value 3.6 Mg hm-2 is taken from Table 1 in Ref. [4] (rounded off from 3591 kg/ha, and the year should 1935, not 1931). The authors should observe that the value 3.6 Mg hm-2 is the average maize yield found in studies done between 1880 and 1960. The mean year of the studies is 1935. This is not to say that the average yield was 3.6 Mg hm-2 in 1935. Similarly, the value 10 Mg hm-2 (rounded off from 10028 kg/ha) is the average maize yield found in studies done between 2006 and 2012. The mean year of the studies is 2010. This is not to say that the average yield was 10 Mg hm-2 in 2010. The authors should note that the numbers are based on research studies – not average farm yields. According to FAOSTAT, the average farm yield for maize in USA was 3918 kg/ha in 1961 and 9576 kg/ha in 2010. (The authors should also observe that there is a fundamental difference between the terms maize yield and maize production.)

Line 36: The authors use Thomas et al. (2002) as a reference (Ref. [5]) to document their claim that the increase in maize production in USA from 3.6 Mg hm-2 in 1931 to 10 Mg hm-2 in 2010 was due to “increased maize greenness”. However, the paper by Thomas et al. mentions neither the word “greenness” nor the issue of maize production in USA. In fact, the paper by Thomas et al. was published eight years before the end of the quoted maize production period, and it deals with stay-green mutants in grasses and legumes.

Author Response

Response to Reviewer 4 Comments

Round 2

Reviewer 4 Report

Well written. Sound science.